# The Genetic Diversity and Population Structure of Different Geographical Populations of Bottle Gourd (*Lagenaria siceraria*) Accessions Based on Genotyping-by-Sequencing

**Rodrigo Contreras-Soto [1],\*** , **Ariel Salvatierra [2]**, **Carlos Maldonado [1]** and **Jacob Mashilo [3]**

[1]   Instituto de Ciencias Agroalimentarias, Animales y Ambientales—ICA3, Universidad de O'Higgins, San Fernando 3070000, Chile; carlos.maldonado@uoh.cl
[2]   Centro de Estudios Avanzados en Fruticultura, Rengo 2940000, Chile; asalvatierra@ceaf.cl
[3]   Limpopo Department of Agriculture and Rural Development, Towoomba Research Station, Bela-Bela 0480, South Africa; jacobmashilo.jm@gmail.com
\*   Correspondence: rodrigo.contreras@uoh.cl

**Abstract:** *Lagenaria siceraria* (Molina) Standl is an important horticultural and medicinal crop grown worldwide in the food and pharmaceutical industries. The crop exhibits extensive phenotypic and genetic variation useful for cultivar development targeting economic traits; however, limited genomic resources are available for effective germplasm characterization into breeding and conservation strategies. This study determined the genetic relationships and population structure in a collection of different accessions of bottle gourd derived from Chile, Asia, and South Africa by using single-nucleotide polymorphism (SNP) markers and mining of simple sequence repeat (SSR) loci derived from genotyping-by-sequencing (GBS) data. The GBS resulted in 12,766 SNPs classified as moderate to highly informative, with a mean polymorphic information content of 0.29. The mean gene diversity of 0.16 indicated a low genetic differentiation of the accessions. Analysis of molecular variance revealed less differentiation between (36%) as compared to within (48%) bottle gourd accessions, suggesting that a random mating system dominates inbreeding. Population structure revealed two genetically differentiated groups comprising South African accessions and an admixed group with accessions of Asian and Chilean origin. The results of SSR loci mining from GBS data should be developed and validated before being used in diverse bottle gourd accessions. The SNPs markers developed in the present study are useful genomic resources in bottle gourd breeding programs for assessing the extent of genetic diversity for effective parental selection and breeding.

**Keywords:** AMOVA; *Lagenaria*; genotyping-by-sequencing; single nucleotide polymorphism; simple sequence repeats

## 1. Introduction

Bottle gourd [*Lagenaria siceraria* (Mol.) Standl., 2*n* = 2x = 22] or calabash is a diploid, monoecious, and cross-pollinating self-fertile vegetable crop belonging to the genus *Lagenaria* of the *Cucurbitaceae* family [1]. The crop is used for diverse and beneficial uses, including food, feed, and medicinal purposes. The fresh and tender fruits are cooked as food and the mature dry fruits are used to make containers for food and grain storage, decorations, and musical instruments [2]. The cultivated bottle gourd is also used as rootstock for production of sweet watermelon (*Citrullus lanatus* var. *lanatus*) to control soil-borne diseases, leaf diseases, and low soil temperature, and to improve nitrogen-use efficiency [3–6] and fruit quality [7,8].

Bottle gourd is thought to be one of the first plant species to be domesticated for human use, approximately 10,000 years ago [9,10]. Archaeological evidence suggested bottle gourd originated in Africa [9] and comprises two subspecies, namely the African *L. siceraria* ssp. *siceraria* and the Asian *L. siceraria* ssp. *asiatica* [11,12]. Although bottle

gourd is native to Africa, the species has been widely grown worldwide due to its abundant genetic and morphological variation which allows adaptation to diverse growing environments [10,12,13]. The cross-pollinating nature of the crop resulted in phenotypic variation for fruit traits, including fruit shape and size [3,14,15], and seed morphology [16]. The fruit and seed characteristics are economic traits in cultivar development of this crop that are important for various domestic and industrial applications.

The extent of genetic diversity in bottle gourd has been previously assessed employing various molecular markers. In India, Sarao et al. [17] fingerprinted 20 accessions of bottle gourd using 20 simple sequence repeat (SSR) markers and reported high genetic diversity among accessions. Mashilo et al. [18] using 11 SSR markers selected distantly related bottle gourd landraces of South Africa origin. Xu et al. [19] using 3226 SNPs markers identified two distinct groups among Chinese bottle gourd accessions based on fruit shape rather than collection site. To date, molecular markers have been used to study population structure and genetic relationships of *L. siceraria*, such as inter-sequence simple repeats [20], SSRs [15,17], and SNPs [19,21]. Next-generation sequencing (NGS)-based SNPs are the most widely used molecular markers to study genome-wide association, population structure, genomic selection, and genetic diversity due to their genome-wide abundance, particularly when many markers are required [22,23]. Genotyping-by-sequencing (GBS) has emerged as one NGS-based genotyping platform for marker design and development [22,23]; in fact, the NGS technology provide large amounts of sequence data to develop numerous SNP and microsatellite markers at the whole-genome scale [24]. Furthermore, this approach provides accurate results independently of the population or target species. Moreover, GBS can produce high marker density without previously available genomic information and can reveal the extent of genetic relatedness and genetic variation within and between cultivated and wild species [22,25].

To date limited genomic resources have been developed for bottle gourd germplasm characterization. This has to some extent limited breeding efforts to determine heterotic groups for the hybrid development, release, and commercialization of bottle gourd cultivars with desired attributes for farmers, consumers, and the food and pharmaceutical industries. Furthermore, quantitative trait loci controlling the expression of key qualitative and quantitative traits remain largely unexplored in bottle gourd, partly owing to the limited development of genomic resources. In the present study we applied GBS, resulting in the development of 12,766 SNPs molecular markers distributed across 11 chromosomes of bottle gourd. Therefore, the purpose of this study was to determine the genetic relationships and population structure in a collection of different accessions of bottle gourd from Chile, Asia, and South Africa using the new-developed SNPs markers and the mining of SSR loci derived from GBS data.

## 2. Materials and Methods

### 2.1. Plant Material

A germplasm collection consisting of 25 bottle gourd accessions originating from different geographic areas of Asia (4), South Africa (15), and South America (6) was used for the current study. Fifteen bottle gourd accessions of South Africa were local varieties grown by farmers in the Limpopo Province of South Africa and sourced from the Limpopo Department of Agriculture and Rural Development (Towoomba Research Station), South Africa. The 4 accessions of Asia were sourced from the Genetic Resource Center of Japan, specifically from the National Agriculture and Food Research Organization (NARO), whereas the accessions of South America were collected and comprised of local populations of Chile and Brazil. Details of the accessions are presented in the Supplementary Material (Table S1).

### 2.2. GBS, Read Clustering, and SNP Calling

Genomic DNA of the 25 accessions was extracted from young leaves collected from 3-week-old seedlings using the QIAGEN DNeasy Plant Mini Kit for DNA extraction (QI-

AGEN, Hilden, Germany) following the manufacturer's instructions. We evaluated the quality of DNA via agarose gel electrophoresis and measured the fluorometric quantification by the Qubit 2.0 and Qubit dsDNA HS Assay Kit (Thermo Fisher Scientific, Waltham, MA, USA). The genotyping-by-sequencing data was generated following the method of Elshire et al. [26]. Briefly, 100 ng of genomic DNA and 3.6 ng of total adapters were used, the genomic DNAs were restricted with ApeKI enzyme, and the library was amplified with 18 PCR cycles. After PCR, the pooled products were purified and quantified for sequencing on the Illumina HiSeq 2000 flow cell for sequencing.

Reads and tags (fastq) found in each sequencing lane from 25 barcodes produced a total of 485 million read pairs and an average of 18.5 million high-quality read-pair counts. The reads for both ends of the pair-end data were combined into individual per-sample files, and aligned to the bottle gourd inbred line USVL1VR-Ls reference genome using bowtie2 [27]. The preset-sensitive end-to-end mapping parameters were used, and the sorted alignments were subsequently used for SNP calling using the Stacks 2.5 pipeline (http://catchenlab.life.illinois.edu/stacks/ (accessed on 18 August 2021)). Alignment and merging resulted in a total of 71,212 called SNPs.

After removing lines with failed data, the GBS data from the 25 accessions were stored in Variant Call Format version 4.1 [28]. Genotyping-by-sequencing datasets typically have high rates of missing data [29]. The linkage disequilibrium k nearest neighbor imputation method [30] was used to impute missing values in this dataset. Only SNPs with a minor allele frequency >0.05% and <25% missing data were filtered, resulting in 12,766 high-quality polymorphic SNPs. The SNP calling was performed using TASSEL version 5.2 in the GBS pipeline [31].

### 2.3. Analysis of Genetic Diversity Parameters and Molecular Variance

Genetic diversity of 25 bottle gourd accessions was analyzed with 12,766 SNPs markers by using the poppr package of R-software [32]. The filtered SNPs were used to calculate genetic diversity parameters such as minor allele frequency (MAF), polymorphic information content (PIC), expected heterozygosity (He), and observed heterozygosity (Ho). These analyses were carried out using the R-package. The PIC value of an l-allele locus was calculated as:

$$1 - \sum_{i=1}^{l} P_i^2 - \sum_{i=1}^{l-1} \sum_{j=i+1}^{l} 2P_i^2 P_j^2 \tag{1}$$

where $P_i$ and $P_j$ are the population frequency of the $i$th and $j$th allele.

Analysis of molecular variance (AMOVA) was carried out by using the poppr package in R to detect population differentiation [33].

### 2.4. Population Structure and Genetic Relationship

The genetic relationship among the landraces of bottle gourd was calculated based on identity-by-state (IBS) distance that represents a kinship matrix using the software TASSEL 5.2 (Trait Analysis by aSSociation, Evolution and Linkage) [34]. The population structure was inferred with the Markov Chain Monte Carlo (MCMC) algorithm for the generalized Bayesian clustering method implemented in Structure software [35]. Consequently, 10 independent runs of MCMC sampling were implemented for numbers of groups (K parameter) varying from 2 to 5. For each run, the initial burn-in period was set to 10,000 with 110,000 MCMC iterations under the non-admixture model and with prior information on the individual's origin. The optimal value of K was estimated from the second-order change rate of the probability function with respect to K (ΔK), as proposed by Evanno et al. [36].

### 2.5. Mining of Simple Sequence Repeats Markers

The Illumina raw reads data were preprocessed to generate clean reads and then analyzed using the core Stack pipeline of Stacks v.2.5 software with default parameters. Each consensus sequence resulting from the Stack pipeline was then screened for simple

sequence repeats (SSRs) using GMATA package with default parameters [37]. The acquired SSRs were considered to only represent those containing perfect repeats of SSRs whose basic motifs ranged from 2 bp to 6 bp with defined minimum repeat units of 5 iterations for di-, tri-, tetra-, penta-, hexa- and heptanucleotide repeats.

## 3. Results

### 3.1. GBS Analysis

Genome sequencing of the 25 bottle gourd accessions using GBS generated a total of 485 million read pairs, with an average read pair count of 18.5 million. Each of the 25 sample reads was mapped to "*Lagenaria siceraria* var. USVL1VR-Ls". In the GBS analysis, a total of 71,212 called and unfiltered SNPs were detected as raw SNP markers. Of these, 12,766 filtered SNPs were obtained and distributed across the 11 chromosomes of *L. siceraria*. The numbers of homozygote and heterozygote SNP loci ranged from 9865 (CLS-013) to 10,594 (CLS-024) with an average of 10,194, and from 2334 (CLS-024) to 3063 (CLS-013) with an average of 2734, respectively (Table 1). The average homozygote rate was approximately 78.9%, and the average heterozygote rate was 21.1% (Figure 1).

**Table 1.** Summary statistics of genetic diversity parameters generated by single nucleotide polymorphic markers across 11 chromosomes of *L. siceraria*.

| Chromosome Number | Number of SNPs Markers | PIC | MAF | Ho | He |
|:---:|:---:|:---:|:---:|:---:|:---:|
| 1 | 1285 | 0.298 | 0.236 | 0.184 | 0.159 |
| 2 | 1580 | 0.297 | 0.238 | 0.208 | 0.161 |
| 3 | 1305 | 0.305 | 0.239 | 0.179 | 0.162 |
| 4 | 1448 | 0.287 | 0.224 | 0.186 | 0.155 |
| 5 | 1272 | 0.292 | 0.228 | 0.177 | 0.157 |
| 6 | 1044 | 0.291 | 0.225 | 0.166 | 0.156 |
| 7 | 855 | 0.286 | 0.222 | 0.176 | 0.154 |
| 8 | 1103 | 0.275 | 0.211 | 0.150 | 0.147 |
| 9 | 1059 | 0.297 | 0.234 | 0.190 | 0.159 |
| 10 | 824 | 0.302 | 0.242 | 0.211 | 0.163 |
| 11 | 991 | 0.295 | 0.231 | 0.172 | 0.158 |
| Total/Average | 12766 | 0.293 | 0.230 | 0.183 | 0.157 |

PIC: polymorphic information content; MAF: minor allele frequency; Ho: observed heterozygosity; HE: expected heterozygosity.

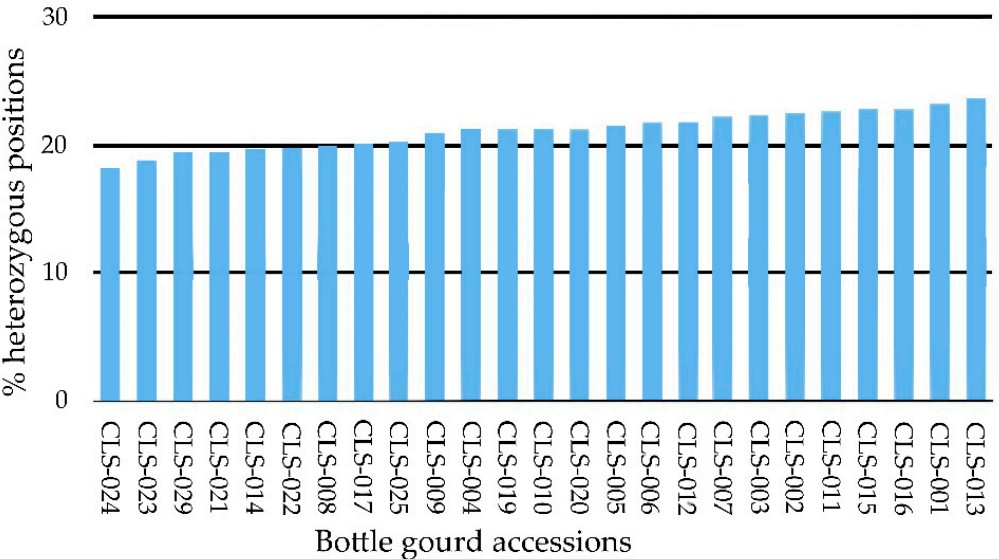

**Figure 1.** Percentage of heterozygous positions of 25 *L. siceraria* accessions of diverse geographical origins generated using single nucleotide markers developed using genotyping-by-sequencing.

The average PIC value across all the markers and chromosomes was 0.26, whereas the observed heterozygosity ranged from 0.15 to 0.22 with an average of 0.18. The expected heterozygosity ranged between 0.15 and 0.16, with a mean of 0.16. Minor allele frequency (MAF) ranged between 0.21 and 0.242, with an average of 0.23. The highest PIC and MAF values were on chromosome 10, whereas the lowest were on chromosome 8 (Table 1).

### 3.2. AMOVA

According to AMOVA, the hypothesis of random mating between the three bottle gourd populations of different geographical origin (Asia, South Africa, and South America) was rejected, with strong evidence that these populations were significantly differentiated at all stratifications (Table 2).

**Table 2.** Analysis of molecular variance among bottle gourd accessions showing percentage of molecular variance explained by each source of variation.

| Component of Differentiation | DF | Mean Square | PVE (%) | Phi Statistics |
|---|---|---|---|---|
| Between populations | 2 | 28,618 | 35.9 | PT = 0.36 |
| Between samples within populations | 22 | 3958 | 16.3 | SP = 0.25 |
| Within samples | 25 | 2357 | 47.8 | ST = 0.52 |
| Total | - | - | 100 | |

DF: degree of freedom; PVE: percentage of variance explained; PT: population-total differentiation; SP: sample-population differentiation; ST: sample-total differentiation.

According to the phi statistics, there was relatively high differentiation between the different levels of comparison. The lowest differentiation was reported among samples within the same population or geographical origin (25%). Substantial differentiation between populations was reported (36%). However, 52% of the differentiation occurred within samples (Table 2).

### 3.3. Population Structure

The population structure and genetic relationship analysis revealed two genetically differentiated groups (Figures 2 and 3). Table 3 shows the results of the statistical parameters that define the number of groups or populations that represented the population structure of the 25 accessions of bottle gourd. Specifically, in Figure 2, the blue color represents the percentage of membership of the 16 accessions of South African origin and the orange color represents the percentage of membership of the 9 accessions of Asian and South American origin. Figure 3 shows the heatmap of the kinship among the 25 accessions of *L. siceraria*. The red and orange color represented the higher relationship within bottle gourd accessions, whereas the yellow color corresponds to a lower relationship.

**Table 3.** Number of individuals by population, and statistics parameters and credible intervals for each cluster of 25 accessions of *L. siceraria*.

| Cluster | Individuals | Mean | Median | Mode * | SD | 95% Credible Interval | |
|---|---|---|---|---|---|---|---|
| | | | | | | Lower | Upper |
| I | 9 | 0.40 | 0.40 | 0.40 | 0.0062 | 0.39 | 0.41 |
| II | 16 | 0.62 | 0.61 | 0.61 | 0.0052 | 0.60 | 0.62 |

* Kernel density estimates of the mode from marginal posterior distributions.

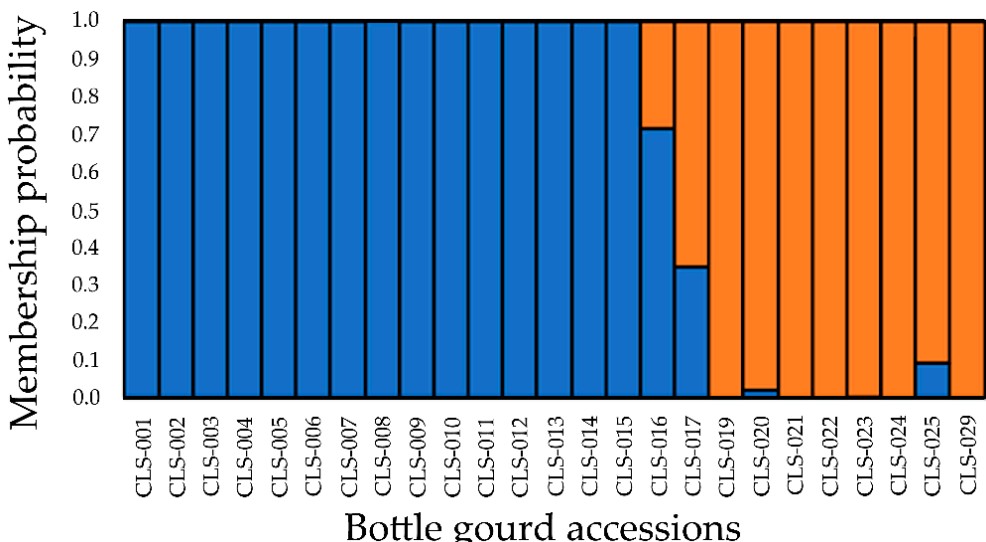

**Figure 2.** Bar plot of the estimated population structure of 25 *L. siceraria* accessions (k = 2). The vertical bars of the *y*-axis represent the estimated membership probability of each population to the subgroups. The blue and orange color represent different subgroups, and the *x*-axis is the bottle gourd accession codes. Populations with colored segments indicates admixed origin.

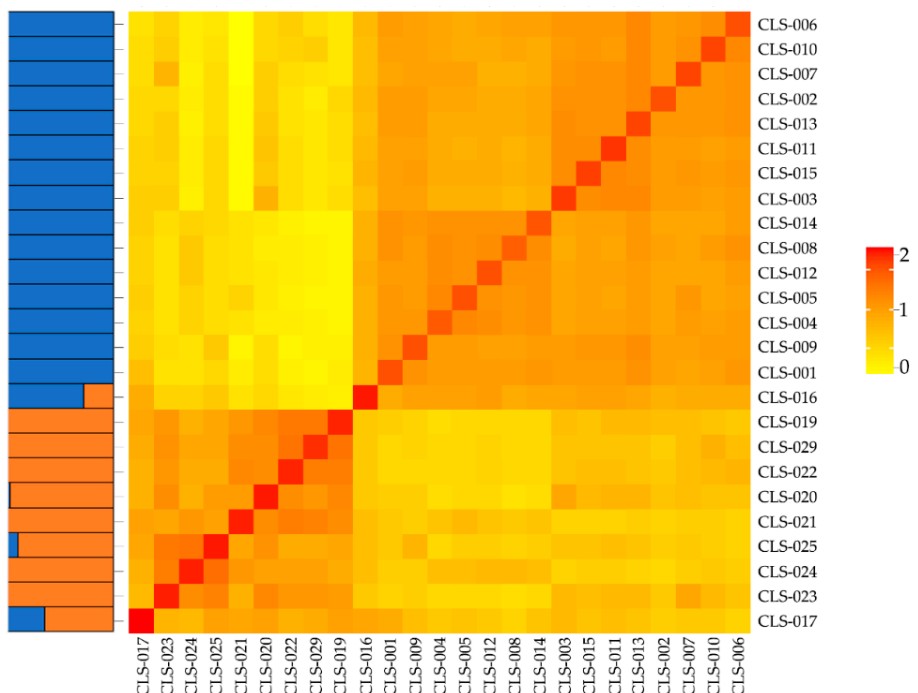

**Figure 3.** Kinship heatmap showing the genetic relationship among the 25 accessions of *L. siceraria* based on 71,212 SNPs markers with two main groups. One group is composed by accessions of South African geographical origin, and the other corresponds to an admixture group with accessions of different geographical origin (Asia and South America). Red and orange colors indicate higher kinship and yellow colors indicate lower kinship. CLS-001 to CLS-015 correspond to accessions of South Africa; CLS-016 and CLS-017 correspond to accessions of Brazil; CLS-019 to CLS-022 correspond to accessions of Asia; and CLS-023, CLS-024, CLS-025, and CLS-029 correspond to accessions of Chile.

### 3.4. Bottle Gourd SSR Locus Identification and the Frequency of SSRs

We used high-quality read pair count sequences derived from GBS data to identify SSR loci in a collection of 25 bottle gourd accessions. The search for SSR-containing regions was restricted to motif of di-, tri-, tetra-, penta-, hexa-, and heptanucleotides. A total of

95,635 SSRs loci with di-, tri-, tetra-, penta-, hexa- and heptanucleotide repeats of 5 or more repeats were identified from the GBS data. These SSR loci consisted of 69,682 dinucleotide repeats (72.86%), 21,641 trinucleotide repeats (22.63%), 3203 tetranucleotides repeats (3.35%), 599 pentanucleotide repeats (0.63%), 356 hexanucleotide repeats (0.37%), and 154 heptanucleotides repeats (0.16%) (Table 4). Dinucleotides and trinucleotides were identified as the most abundant SSR class, representing the 95.49% of the SSR motif classes. The repeat motif AT/AT (26,274) was the most frequent in the dinucleotide SSR, representing 37.71% of the total dinucleotides, and the repeat motif AAT/ATT (7592) was the most frequent into the trinucleotide SSR, representing 31.08% of the trinucleotides (Figure 4).

**Table 4.** Summary of bottle gourd SSRs identified based on GBS sequences.

| SSR Motifs | Number of Repeat Units of Each SSR Motif | | | | | | | | Frequency (%) |
|---|---|---|---|---|---|---|---|---|---|
| | 5 | 6 | 7 | 8 | 9 | 10 | >10 | Total | |
| Dinucleotide | 28,400 | 10,526 | 6872 | 4695 | 3539 | 2679 | 12,971 | 69,682 | 72.86 |
| Trinucleotide | 9670 | 4589 | 2399 | 1386 | 855 | 603 | 2139 | 21,641 | 22.63 |
| Tetranucleotide | 2372 | 537 | 160 | 68 | 28 | 12 | 26 | 3203 | 3.35 |
| Pentanucleotide | 424 | 125 | 33 | 10 | 1 | 3 | 3 | 599 | 0.63 |
| Hexanucleotide | 250 | 82 | 16 | 4 | 2 | 1 | 1 | 356 | 0.37 |
| Heptanucleotide | 105 | 20 | 15 | 2 | 1 | 2 | 9 | 154 | 0.16 |
| Total | 41,221 | 15,879 | 9495 | 6165 | 4426 | 3300 | 15,149 | 95,635 | 100.00 |

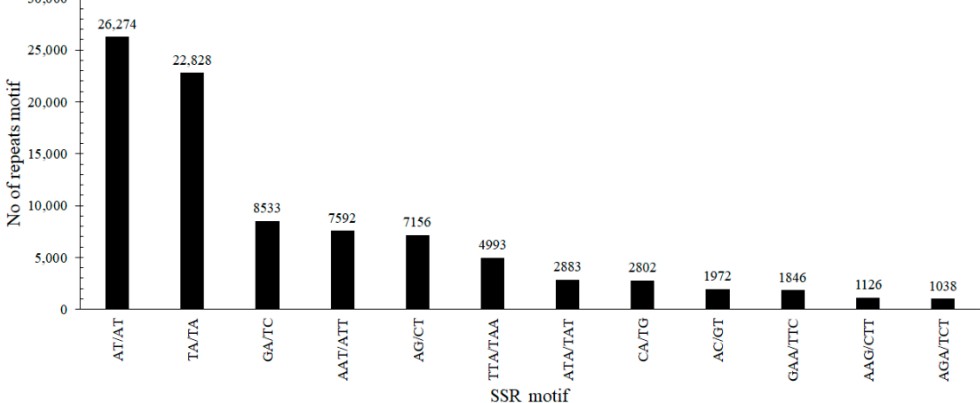

**Figure 4.** Frequency distribution of the 6 most frequent dinucleotide and trinucleotide SSR motifs. Only the 6 most frequent nucleotide motifs are shown.

## 4. Discussion

Effective use of bottle gourd genetic resources for cultivar development and conservation requires development of genomic tools for marker-assisted breeding. During the last decade, significant progress has been made in the development of genomic resources in bottle gourd. Most of these genomic resources provide valuable information about genetic relationships among genotypes for effective selection and use in breeding programs [19,27,38]. Despite significant progress, there are generally very limited genomic resources developed for bottle gourd limiting breeding efforts to develop competitive genotypes for agricultural production and in the nutraceutical and pharmaceutical industries. The present study identified SNP molecular markers distributed across 11 chromosomes of bottle gourd by employing a genotyping-by-sequencing platform. These were then used to determine genetic relationships and population structure in a collection of bottle gourd accessions of African, Asian, and South American origin, and SSR loci were subsequently identified from the GBS sequences. Among the high-throughput sequencing technologies, GBS is considered the most cost-effective tool to identify and genotype many polymorphisms at the genome scale [27]. Here, we used Elshire-GBS method and *L. siceraria* var. USVL1VR-Ls as reference genome which resulted in a set of 12,766 filtered SNP markers. A recent GBS study used to confirm the varietal status of bottle gourd accessions pro-

duced 22,575 SNPs [21], which was higher than the present study. Other high-throughput studies conducted in bottle gourd used restriction site-associated DNA sequencing (RAD-Seq), a form of GBS that generates low coverage genome sequencing in which reference genomes are not available [19,27]. In addition, Wu et al. [27] using RAD-Seq and aligning to the Hangzhou gourd reference genome detected 19,226 SNPs, similar to the present findings. On the contrary, Xu et al. [19] using RAD-Seq genotyping identified 3226 SNPs and Xu et al. [39] using partial sequencing only discovered 3913 putative SNPs. These differences between the current study and previous results may be due to high read depth variation of RAD-Seq or the high levels of missing data of Elshire-GBS [40] and the average coverage which typically varies between these reduced-representation sequencing methods. For instance, while RAD-seq involves sequencing fragments to moderate coverage between $5\times$ and $15\times$ [41], Elshire-GBS studies tend to reach low coverage of $\sim 1\times$ [42]. Despite these differences, the generated SNP markers and SSR loci are a useful genomic resource for genetic analysis and breeding in bottle gourd for diverse applications, however, in subsequent studies, a final set of SSR loci should be developed and validated before being used in diverse bottle gourd accessions collected from different regions of the world.

For instance, in this study, the most abundant class of SSRs identified from GBS sequences was comprised by dinucleotide and trinucleotide repeats. Similar results have been reported previously for bottle gourd. Xu et al. [39], for example, identified that dinucleotide and trinucleotide repeats were the most abundant, while mononucleotide and pentanucleotide repeats were relatively rare. Moreover, the high frequency of dinucleotide and trinucleotide repeats is consistent with other cucurbit species, including cucumber and watermelon [43,44]. Furthermore, similar to our results, the AT-rich motifs have been the predominant motif in all nucleotide repeats in melon, watermelon, cucumber, and bottle gourd genomes [24,43,44].

In a breeding program, the extent of genetic diversity and population relationships among the germplasm is useful to identify distantly related parents for hybridization to develop genetically improved genotypes of bottle gourd for rootstocks, food, feed, and for medicinal purposes. For this reason, in different regions, several studies have been conducted to determine the genetic diversity of bottle gourd accessions [15,45,46]. In this study, the accessions of bottle gourd were collected from Chile, Japan (Philippines, South Korea), and South Africa. For instance, the SNP set was not able to distinguish between var *hispida* represented by the Asian accessions CLS-19, CLS-21, and CLS-22 and var *siceraria* represented by CLS-020. On the other hand, most of the Asian accessions share similar genetic background to South African accessions which have been previously assayed using SSR markers [47]. In the current study, various genetic parameters were estimated using SNPs markers including Ho, He and PIC values with mean values of 0.18, 0.16 and 0.29, respectively. Gürcan et al. [45], genotyped thirty-one bottle gourd accessions from USA, India, Nigeria and Russia using SSR markers and reported mean values of 0.50, 0.13, and 0.50 for He, Ho and PIC, in that order. Furthermore, Mashilo et al. [15] using SSR markers reported high average values for He = 0.657 and PIC = 0.57 among bottle gourd accessions, higher than values reported in the present study. Botstein et al. [48] classified the PIC values into three categories (1) if the PIC value of the marker is more than 0.5, the marker is considered highly informative, (2) if the PIC value ranged from 0.25 to 0.5, the marker is a moderately informative, and (3) if the PIC value less than 0.25, then the marker is slightly informative. Based on Botstein classification, SNPs markers generated in the present study are moderately informative. A recent study indicated that PIC values calculated with SNPs markers showed the lowest values compared to SSR markers [49,50]. This can be attributed to the bi-allelic nature of the SNPs which is restricted to PIC values ranging from 0.0 to 0.5 (i.e., when the two alleles have identical frequencies), whereas for SSR markers which are multi-allelic PIC value can vary between 0.5 and 1.0 [49,51].

Expected heterozygosity is usually preferred to assess genetic diversity because it is less sensitive to the sample size than the observed heterozygosity [52]. According to Chesnokov and Artemyeva [52], when Ho and He are similar (i.e., not significantly

different), the crossing in the population is almost accidental. When Ho < He, it is an inbred population, and when Ho > He, the random mating system dominates inbreeding in the population. Our results showed that Ho was slightly higher than He, suggesting that random mating system dominates inbreeding in the assessed bottle gourd germplasm. Moreover, population differentiation indicated a higher variation within sample, a common characteristic of cross-pollinated plants which can reduce the loss of genetic diversity through large gene flow. As proposed by Mashilo et al. [15], this could be attributed to the high out-crossing nature of bottle gourd or long-term selection of the crop by farmers for diverse uses.

Population structure and genetic relatedness are useful to understand genetic diversity, differentiate the population according to their geographical origin and conduct association mapping studies. Based on population structure analysis, two genetically differentiated groups were identified; the first including all the accessions originated from South Africa and the second group comprising of Asian and Chilean accessions. These results agree with previous studies conducted in bottle gourd, which reported that clustering of different landraces was independent of geographical location [15,17,45,53]. Another explanation is that the founder effect followed by artificial selection based on fruit shape which tend to generate high genetic similarity [39,54]. In crop improvement programs, germplasm collection missions should be based on morphological variation rather than geographical origin [15]. Heiser [55] classified bottle gourd into two subspecies: Asian and American African subspecies. These authors postulated that African wild bottle gourd floated to the shores of America and were independently domesticated there. This may explain why the accession with provenance of Brazil (CLS-016) was clustered with the African accessions; however, further studies should be conducted in this regard. Using various molecular markers, different results on the phenomenon have been reported. For example, Erickson et al. [10] using SNP markers within chloroplast DNA concluded that American bottle gourds were more closely related to Asian than to African gourds, whereas Decker-Walters and Wilkins-Ellert [9] by using RAPD molecular markers revealed that American germplasm is distinct and primarily originated from Africa but possesses Asian genetic profiles. Similarly to Erickson et al. [10], our results supported the idea that one group is only composed by accessions of South Africa, and the other correspond to an admixture group with accessions from Asia and South America.

## 5. Conclusions

The present study genotyped bottle gourd accessions of diverse origins using new-developed single nucleotide polymorphism markers. A total of 12,766 SNPs molecular markers were generated using genotyping-by-sequencing and were classified as moderate to highly informative. Low genetic differentiation was observed among the assessed bottle gourd accessions using SNPs markers. The random mating system was found to dominate inbreeding in the assayed bottle gourd population. Accordingly, two genetically differentiated groups comprising of South African accessions and an admixed group with accessions of Asian and Chilean origin were identified. The results of SSR loci mining from GBS data should be developed and validated before being used in diverse bottle gourd accessions. The SNPs developed in the present study is a useful genomic resource for bottle gourd breeding targeting development of genetically improved genotypes for diverse uses, including rootstocks, food, feed, and medicine.

**Supplementary Materials:** The following are available online at https://www.mdpi.com/article/10.3390/agronomy11081677/s1, Table S1: Accession code, origin, region, location, and geographical coordinates of 25 bottle gourd accessions used at the present study.

**Author Contributions:** Conceptualization, Methodology, Resources: R.C.-S.; Writing draft and editing: R.C.-S., A.S., C.M. and J.M.; Data analysis: R.C.-S. and C.M.; Experimental work: R.C.-S. and A.S. All authors have read and agreed to the published version of the manuscript.

**Funding:** This research was funded by the National Commission for Scientific and Technological Research (CONICYT, Chile) FONDECYT, grant number 11180278.

**Institutional Review Board Statement:** Not applicable.

**Informed Consent Statement:** Not applicable.

**Acknowledgments:** R.C.-S. thanks the Centro de Estudios Avanzados en Fruticultura (CEAF) for support of the project, the National Agricultural and Food Research Organization (NARO), and the Limpopo Department of Agriculture and Rural Development of South Africa for providing plant material.

**Conflicts of Interest:** The authors declare no conflict of interest.

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
