# Peer review of "The Genetic Diversity and Population Structure of Different Geographical Populations of Bottle Gourd (Lagenaria siceraria) Accessions Based on Genotyping-by-Sequencing"

_agronomy, doi:10.3390/agronomy11081677_

Round 1
Reviewer 1 Report
In this manuscript entitled "Genetic diversity and population structure of different geo-2 graphical populations of bottle gourd (Lagenaria siceraria) accessions based on Genotyping-By-Sequencing" the authors investigated basic genetic parameters of 25 bottle gourd accessions and developed amount of SSR markers for future studies. I believe that the manuscript is solid and has been very well worked and provides all the necessary information for its understanding. Even though the manuscript is remarkable and worth publishing, there are a couple of details that could be easily improved.
Comments:
- A lots of SSR markers have been identified in this study. But, the basic genetic parameters of 25 bottle gourd accessions were just calculated on SNPs markers. Is it different between two types molecular markers were used in estimation of basic genetic parameters?
- How about the distribution of SSR markers in whole genome?
- Figure5 has no real meaning. It should be used for performance SSR type with high diversity.
Author Response
In this manuscript entitled "Genetic diversity and population structure of different geo-2 graphical populations of bottle gourd (Lagenaria siceraria) accessions based on Genotyping-By-Sequencing" the authors investigated basic genetic parameters of 25 bottle gourd accessions and developed amount of SSR markers for future studies. I believe that the manuscript is solid and has been very well worked and provides all the necessary information for its understanding. Even though the manuscript is remarkable and worth publishing, there are a couple of details that could be easily improved.
Thank you for reviewing our manuscript and providing valuable feedback. All changes have been highlighted in red in the text.
- A lots of SSR markers have been identified in this study. But, the basic genetic parameters of 25 bottle gourd accessions were just calculated on SNPs markers. Is it different between two types molecular markers were used in estimation of basic genetic parameters?
Dear reviewer, this is a good question. The values of estimation of these genetic parameters calculated with both markers should be similar, but not the same. On the other hand, in this manuscript our focus was not to compare the differences between the estimation of the genetic parameters with SNPs and SSR markers. In addition, we propose to validate the SSR markers.
- How about the distribution of SSR markers in whole genome?
In this table we incorporate a resume with the occurrence of motifs and SSR frequency distribution in L. siceraria genome.
|
Chromosome |
Total Motifs |
Frequency (Motifs/Mbp) |
|
1 |
1233 |
216.90 |
|
2 |
8925 |
303.74 |
|
3 |
11709 |
323.85 |
|
4 |
10061 |
297.03 |
|
5 |
9823 |
311.70 |
|
6 |
9286 |
320.08 |
|
7 |
7617 |
280.68 |
|
8 |
6905 |
310.30 |
|
9 |
7462 |
303.18 |
|
10 |
8879 |
314.38 |
|
11 |
6137 |
297.95 |
|
12 |
7627 |
300.90 |
|
Total |
95664 |
3580.69 |
|
Average |
7972 |
298.39 |
- Figure 5 has no real meaning. It should be used for performance SSR type with high diversity.
Thanks for your comment. In your opinion ¿its necessary to eliminate it?

Reviewer 2 Report
The language should be improved for smoother reading, a few examples:
p1/13 ...grown worldwide [serving | for] food and pharmaceutical industries
p1/13 The crop exhibits extensive...
p1/17 accessions of bottle gourd prevenient [provenient? derived?] from Chile
An overstatement (or imprecise language?) is
p1/16 to 18 determined the genetic relationships and population structure...by using single nucleotide polymorphism (SNPs) markers and mining of simple sequence repeats (SSR) loci
the mining of SSR loci from gbs data does not determine anything, only there application would do so, as the authors correctly state a few lines later (p1/25)
p2/45 originated in Africa [9] and compriseds of two subspecies
p2/79 we developed GBS
should read as "we applied GBS"
factual errors in the Materials and Methods section:
p3/105 Elshire et al. [26] method and included the following changes:
The following details (100ng gDNA, 3.6ng adapters, ApeKI restriction enzyme, 18 PCR cycles) are exactly as described by Elshire et al.
p3/110 found in each sequencing lane from 96 barcodes
the study deals only with 25 gourd accessions, so I assume only 25 barcodes were used to allow for simultaneous sequencing of these samples in one lane?
Design of the study, discussion of the results:
A weak point is the inclusion of only 25 accessions in the gbs experiment, and of these only sparse information is given in the supplement table. eg there are four and five accessions from two origins in South Africa included, and of these we don't know if there is some differentiation in fruit or seed characteristics, if they have been grown next to each other for their different qualities, or have they just been collected into the same seed bank? Another finding the authors did not discuss is that the SNP set is not able to distinguish between var hispida and var siceraria (Fig 4, CLS-19,21,22 hispida, CLS-020 siceraria). Also one of the accessions from Brazil (CLS-016) clusters with the African accessions, which is not discussed. This information on the geographic origin should be incorporated in Fig 4 for better interpretation of the findings. The dendrogram on top of Fig 4 is difficult to read, I would have a larger representation, and have only one figure of the membership probabiliy, which is doubled between Fig 3 and Fig 4. Fig 2 as graphic reprentation of transition/transversion has little information beyond the table, so I would remove it.
Author Response
Dear reviewer.
Thank you for reviewing our manuscript and providing valuable feedback. All changes have been highlighted in red in the text.
The language should be improved for smoother reading, a few examples:
We improved several sentences according with your suggestions.
p1/13 ...grown worldwide [serving | for] food and pharmaceutical industries
It was modified as suggested, please see line 13.
p1/13 The crop exhibits extensive...
It was modified as suggested, please see line 13.
p1/17 accessions of bottle gourd prevenient [provenient? derived?] from Chile
It was modified as suggested, please see line 17.
An overstatement (or imprecise language?) is
p1/16 to 18 determined the genetic relationships and population structure...by using single nucleotide polymorphism (SNPs) markers and mining of simple sequence repeats (SSR) loci the mining of SSR loci from gbs data does not determine anything, only there application would do so, as the authors correctly state a few lines later (p1/25)
Many thanks for your comment.
p2/45 originated in Africa [9] and compriseds of two subspecies
It was modified as suggested, please see line 45.
p2/79 we developed GBS
should read as "we applied GBS"
It was modified as suggested, please see line 79.
factual errors in the Materials and Methods section:
p3/105 Elshire et al. [26] method and included the following changes:
The following details (100ng gDNA, 3.6ng adapters, ApeKI restriction enzyme, 18 PCR cycles) are exactly as described by Elshire et al.
To clarify, we modified to evict mis-confusion. Please see line 105.
p3/110 found in each sequencing lane from 96 barcodes
the study deals only with 25 gourd accessions, so I assume only 25 barcodes were used to allow for simultaneous sequencing of these samples in one lane?
Yes, this is correct. Only 25 barcodes were used. Many thank for your comment. We modified in the text (Please see line 109).
Design of the study, discussion of the results:
A weak point is the inclusion of only 25 accessions in the gbs experiment, and of these only sparse information is given in the supplement table. eg there are four and five accessions from two origins in South Africa included, and of these we don't know if there is some differentiation in fruit or seed characteristics, if they have been grown next to each other for their different qualities, or have they just been collected into the same seed bank?
We incorporate more information in supplementary material Table S1. In addition, if necessary, we can incorporate the image (included at the final of this document - Page 3) in which is showed the phenotypic fruit differences among the accessions.
In Table S1, We incorporate the information that the accessions of South Africa showed differentiation in fruit characteristics as previously informed by Mashilo et al. (2017).
Mashilo, J.; Shimelis, H.; Odindo, A. Phenotypic and genotypic characterization of bottle gourd [Lagenaria siceraria (Molina) Standl.] and implications for breeding: A Review. Sci. Hortic. 2017, 222, 136–144. doi: 10.1016/j.scienta.2017.05.020.
The seeds were recollected from different origins as showed in the Table S1, but these were supported by the same seed bank.
Another finding the authors did not discuss is that the SNP set is not able to distinguish between var hispida and var siceraria (Fig 4, CLS-19,21,22 hispida, CLS-020 siceraria).
We added more information in Discussion (Please see lines: 300-302).
Also one of the accessions from Brazil (CLS-016) clusters with the African accessions, which is not discussed. This information on the geographic origin should be incorporated in Fig 4 for better interpretation of the findings.
We added more information in Discussion (Please see lines: 344-346).
We incorporate information of the geographic origin in the caption of the Figure 3 (Please see lines 224-226).
The dendrogram on top of Fig 4 is difficult to read, I would have a larger representation, and have only one figure of the membership probabiliy, which is doubled between Fig 3 and Fig 4.
To improve this figure, we eliminate the dendrogram on top of Figure 3.
Fig 2 as graphic reprentation of transition/transversion has little information beyond the table, so I would remove it.
According with your comment, we decide to eliminate it. The numeration of the figures were modified.
Phenotypes of Lagenaria siceraria fruits. Representative fruits of genotypes a)CLS-001, b) CLS-002, c) CLS-003, d) CLS-004, e) CLS-005, f) CLS-006, g) CLS-007, h) CLS-008, i) CLS-009, j) CLS-010, k) CLS-011, l) CLS-012, m) CLS-013, n) CLS-014, o) CLS-015, p) CLS-016, q) CLS-017, r) CLS-019, s) CLS-020, t) CLS-021, u) CLS-022, v) CLS-023, w) CLS-024, x) CLS-025, y) CLS-029
